# The Production of Antibiotics Must Be Reoriented: Repositioning Old Narrow-Spectrum Antibiotics, Developing New Microbiome-Sparing Antibiotics

**DOI:** 10.3390/antibiotics11070924

**Published:** 2022-07-08

**Authors:** Sylvain Diamantis, Nicolas Retur, Benjamin Bertrand, Florence Lieutier-Colas, Philippe Carenco, Véronique Mondain

**Affiliations:** 1Groupe Hospitalier Sud Ile de France, 77000 Melun, France; 2EA 7380 Dynamic, Université Paris Est Créteil, EnvA, USC ANSES, 94010 Créteil, France; 3Pôle Pharmacie-Stérilisation—Archet Hospital, Nice University Hospital, 06200 Nice, France; retur.n@chu-nice.fr; 4Hospital Pharmacy—Grasse Hospital, 06130 Grasse, France; benjamin.bertrand@gmail.com; 5AntibioEst, CPIAS Grand Est, de Brabois Hospital, Nancy University Hospital, 54511 Vandoeuvre-les-Nancy, France; f.lieutier@chru-nancy.fr; 6Infection Control Unit, Archet Hospital, Nice University Hospital, 06200 Nice, France; pcarenco@ch-hyeres.fr; 7CPIAS PACA Hôpital Sainte Marguerite, 13009 Marseille, France; 8Infectious Disease Department, Archet Hospital, Nice University Hospital, 06200 Nice, France; mondain.v@chu-nice.fr

**Keywords:** antibacterial, narrow-spectrum antibiotics, research and development, antibiotics, ecological impact

## Abstract

The development of broad-spectrum antibiotics to control multidrug-resistant bacteria is an outdated business model. This strategy has led to the introduction of highly effective antibiotics, but their widespread use has contributed to the emergence of even broader antibiotic resistance. In a strategy to combat antimicrobial resistance, we believe that the use of narrow-spectrum antibiotics should be promoted. This should involve both the repositioning of old antibiotics and the reorientation of research and development towards new narrow-spectrum antibiotics with a low ecological impact. These antibiotics could be prescribed for common conditions such as sore throats and cystitis, which account for the bulk of antibiotic use in humans. Narrow-spectrum, targeted, microbiome-sparing antibiotics could help control antibiotic resistance while being economically sustainable. Their development and production should be supported by governments, which would ultimately benefit from reduced health care costs.

## 1. Introduction and Literature Review

The annual number of deaths due to antibiotic resistance is estimated at 33,000 in Europe and 35,000 in the USA [1,2]. Globally, there are at least 700,000 deaths due to drug-resistant infections each year; O’neil estimates that this figure will exceed 10 million deaths per year by 2050 [2,3].

The ability of bacteria to become resistant to antibiotics is a systematic process directly related to antibiotic use [4]. The unnecessary use of antibiotics accelerates the development and spread of resistance [5]. The use of broad-spectrum antibiotics has an ecological impact that favours the emergence of resistance [6]. To curb the emergence of antibiotic resistance, it is essential to restrict the use of antibiotics with a high ecological impact [7].

Most new antibiotics in the pipeline are broad-spectrum compounds with a high ecological impact, driving up resistance rates [8,9]. To counteract this unintended consequence, antimicrobial stewardship was promoted, driving down overall antimicrobial consumption [10]. This in turn led to a decrease in the profits generated by pharmaceutical companies and to disinvestment in research for new antibiotics [11,12,13].

To avoid the threat of a post-antibiotic era, we believe this vicious circle must be broken.

Our objective in this viewpoint is to promote our vision of the means to confront the dearth of new antibiotics and to call for a paradigm shift in antibiotic research, by focusing on antibiotics with a low ecological impact and a narrow spectrum that can be used extensively. After providing our interpretation of the history of antibiotic production in Section 1.1, the issues related to the development of broad-spectrum antibiotics and the problem of antibiotic resistance in Section 1.2 and Section 1.3, we will put forward proposals to reposition the various antimicrobials and define our view of what should be done in terms of research and production.

### 1.1. A Brief History of Antibiotics and Antibiotic Therapy

Modern research in antibiotics began with Paul Ehrlich and accelerated after World War II with a Golden Age during the 1950s and the 1960s [14,15]. It was followed by a decline in the discovery of new compounds [16]; therefore, most of the major pharmaceutical companies shut down their antibiotic R&D departments due to the belief that all the “low-hanging fruit” had been harvested [17]. On the other hand, investments aimed at discovering new synthetic antibiotics have proved unsuccessful except for a few cases. During the Golden Age, there was little incentive to curtail the use of antibiotics even when not needed as there were plenty of new antibiotics in the pipeline to offset the selection of resistance. Still today, a lot of unnecessary antibiotics are prescribed in outpatient care to patients with respiratory viruses [18,19]. 

Since the 2000s, bacterial resistance has increased rapidly with the emergence of new resistance enzymes spreading in hospitals but also among the general population, namely plasmid-mediated cefotaximases among extended-spectrum beta-lactamase-producing bacteria (CTXM ESBLs) and then carbapenemases with New Delhi metallo-beta-lactamases (NDMs) in India and Oxacillinase-48 (Ox48s) in Africa [20,21,22,23]. The morbidity, mortality and additional costs associated with antibiotic resistance are now well-known, and many authors predict an imminent “post-antibiotic era” [2,24]. In the absence of new molecules to deal with the emergence of resistance, strategies to control antibiotic consumption have been implemented in both outpatient care and in hospitals through the promotion of antibiotic stewardship [25].

### 1.2. The Development of Wide-Spectrum Antibiotics with the Aim of Targeting Multi-Resistant Bacteria Is an Obsolete Business Model 

One of the main strategies has been to restrict the use of classes of molecules identified as particularly resistance-generating (carbapenems and fluoroquinolones) [7,26]. Another strategy has been to classify molecules into routine, monitored and reserve antibiotics [2]. Reserve molecules are those molecules that retain activity on highly resistant bacteria. Thus, the de-escalation strategy, consisting in switching molecules following bacterial identification within 72 h of treatment initiation and shortening treatment duration, reduces the use of compounds that cause resistance and, consequently, the return on investment for the pharmaceutical industry [27,28]. In 2020, 46 new antibiotics were in development, including 26 antimicrobial agents active against WHO priority pathogens, 10 of which were β-lactam and β-lactamase inhibitor combinations [29]. The list of priority pathogens was defined by a consortium of international experts [30]. The development of new antibiotics is mainly aimed at bacteria grouped under the ESKAPE acronym (Enterococcus faecium, Staphylococcus aureus, Klebsiella pneumoniae, Acinetobacter baumannii, Pseudomonas aeruginosa and Enterobacter spp.) [9,30]. These bacteria are indeed characterized by the high frequency of antibiotic resistance as well as their relatively common occurrence in inpatient care settings, especially in hospitals [2]. In North America, resistance of Gram-positive bacteria is a major problem: 80% of E. faecium isolates are resistant to glycopeptides and more than half of S. aureus isolates are resistant to methicillin (MRSA). However, these rates vary greatly between countries, and the percentage of MRSA- and vancomycin-resistant enterococci (VRE) is less than 1% in the Netherlands [31]. 

This list, which was established to respond to the problem of highly resistant bacteria, has unfortunately only addressed the issue of the fight against antibiotic resistance from a therapeutic rather than a preventive perspective. The adage “prevention is better than cure” has been forgotten, and the development of antibiotics with a low ecological impact has not been considered. 

The increase in resistance rates seems to create new market opportunities for antibiotics. The expected number of sales is fundamental for the pharmaceutical industry because clinical development of antibiotics is costly and risky. Only 20% of antimicrobials entering phase one will ever be marketed for medical use. The commercialization of an antibiotic only becomes profitable for a laboratory shortly before the loss of its licence. Antibiotics have been produced specifically to treat infections caused by highly resistant bacteria, but the absolute number of infections caused by highly resistant bacteria is relatively low in countries able to purchase these new antibiotics. As soon as the new broad-spectrum antibiotics become available, antibiotics stewardship policies will restrict the prescription of these molecules: firstly, by limiting prescriptions to documented infections with highly resistant bacteria [7] and, secondly, by promoting short treatment durations [28,32]. 

Each newly approved antibiotic thus captures a smaller and smaller share of the market for shortened treatment durations and thus achieves very disappointing sales [15]. 

### 1.3. Financial and Institutional Measures Have Allowed the Introduction of New Broad-Spectrum Molecules but Are Not a Sustainable Strategy in the Medium Term 

In view of this situation, governments have implemented strategies to support the production of antibiotics by the pharmaceutical industry. Two strategies are in place: “pushes” to propel new drug candidates produced by small companies through funding and special approvals; and “pulls” to support the next step after marketing approval [33,34,35]. 

These strategies have had many favourable effects with a significant increase in new antibiotics being marketed. Unfortunately, only about ten molecules in development today are microbiome-sparing antibiotics. On the other hand, most new antibiotics target the same resistant bacteria (carbapenem-resistant enterobacteria). The competition generated by this phenomenon, also known as “me too”, has contributed to disappointing sales by fragmenting the market [36]. The firm Achaogen filed for bankruptcy in 2019 despite the marketing approval of plazomicin [37]. 

## 2. Methodology

We reviewed published papers focusing on antibiotic production and antibiotic history. MEDLINE was searched through PubMed with June 2022 as the publication date limit, using MESH terms. The MESH terms included «antibiotic access»; «antibiotic innovation»; «financial incentives»; «antimicrobial resistance»; «narrow-spectrum antibiotics»; «ecological impact».

## 3. Results

### 3.1. How to Reposition the Different Antibiotics and Define Research and Production Needs

Antibiotics can be classified according to their spectrum (narrow or broad), their ability to remain active against highly resistant bacteria or their ability to spare the patient’s microbiota. Public health authorities, researchers and industry need to work together to define and meet the needs through a four-fold approach. 

#### 3.1.1. Producing New Drugs 

Funders, governments and health insurers should establish target product profiles according to their needs and specific epidemiological characteristics for the production of new antibiotics. These should fall into two distinct categories: (1) broad-spectrum antibiotics designed to be effective against highly resistant bacteria with low anticipated sales (niche products), and (2) “microbiome-sparing antibiotics” with high anticipated use. 

#### 3.1.2. Repositioning Old Antibiotics

Old molecules can be repositioned by proposing new indications that require innovation in prescribing methods. For example, temocillin was forgotten for many years, then remarketed and positioned as an alternative to carbapenems in infections due to 3GC-resistant Enterobacteriaceae [38]. The short half-life and high MICs require optimization of PKPD parameters to achieve target concentrations by using continuous infusion [39,40]. This repositioning process should be initiated by clinicians and subsequently validated by researchers. 

#### 3.1.3. Promoting the Use of Old Narrow-Spectrum Antibiotics 

Many old narrow-spectrum antibiotics can be positioned in good practice guidelines based on existing literature or recommendations. For example, penicillin V used in the treatment of strep throat in Scandinavian countries could be included in other national guidelines [41]. Another example is the positioning of temocillin in the guidelines published by the Infectious Diseases Society of America and the European Society of Clinical Microbiology and Infectious Diseases as an alternative to carbapenems in acute pyelonephritis [42,43]. This effort should be led by scientific societies and regulatory health authorities. 

### 3.2. Narrow-Spectrum Antibiotics with “Targeted Microbiome-Sparing Antibiotics” Could Help Control Antibiotic Resistance while Remaining Economically Sustainable

In recent years, the concept of the ecological impact of antibiotics, described 20 years ago, seems to have undergone a revival. The evaluation of the ecological impact of antibiotics in patients is defined by A. Andremont et al. as follows: “it encompasses the emergence and spread of resistance genes and resistant strains as well as changes in the distribution of microbial populations in the human commensal flora or environmental flora” [6]. 

All new antibiotics on the market have been designed to address the problem of multidrug resistance without any consideration for their potential ecological impact. Multidrug resistance is in many situations a marginal issue with respect to all prescribed antibiotic therapies. However, these therapies contribute to the emergence of multidrug resistance because of their broad spectrum and their ecological impact. Following the logic of “prevention is better than cure”, we propose to develop narrow-spectrum molecules with a low ecological impact. This strategy would drastically reduce the selection pressure on commensal flora and contain the emergence of resistance. On the other hand, the narrow spectrum of the antibiotic limits its use to a precise indication; for example, bacterial cellulitis can be treated with penicillin G [44]. Penicillin G has a very narrow spectrum and thus a lower ecological impact than molecules such as ertapenem or tigecycline, which have marketing authorizations for this indication. 

This concept, introducing the notion of “targeted microbiome-sparing antibiotics”, has recently been supported by Alm in 2020 [45] and Avis T et al. in 2021 [46]. Older molecules such as penicillin G or temocillin can now be reused and optimized by using continuous infusions [39,47]. The pharmaceutical industry must engage in the production of new molecules with an even narrower spectrum and a lower ecological impact. 

More than a dozen new narrow-spectrum antibiotics are currently being developed. For example, murepavadine (POL7080) is an inhibitor of LptD, a membrane protein involved in surface lipoprotein (SLP) transport, with a very narrow spectrum restricted to Pseudomonas aeruginosa [48]. Afabicin (DEBIO-1450) is an inhibitor of Fab1 involved in the synthesis of membrane fatty acids with a very narrow activity directed exclusively against Staphylococcus aureus [49]. 

It would then be possible to prescribe narrow-spectrum antibiotics that spare the individual’s microbiota. The almost exclusive use of narrow-spectrum molecules would require a significant shift in antibiotic prescription patterns. 

The systematic use of broad-spectrum antibiotics in the probabilistic treatment of infections has atrophied our diagnostic skills. Thus, a third-generation cephalosporin will be equally effective on a community skin, lung or digestive infection. Broad-spectrum antibiotics have not stimulated our clinical ability to distinguish viral infection from bacterial infection. Nor has the use of broad-spectrum antibiotics fostered the search for and use of rapid microbiological diagnostic tools to identify the pathogen and de-escalate to narrower-spectrum antibiotics. 

The use of narrow-spectrum antibiotics requires a precise diagnosis, supported by modern and rapid biological methods, and their use would recompose the set of recommendations for antibiotic therapy. In general practice, many microbial infections involve only a narrow array of bacteria. For example, strep throat, streptococcal cellulitis and all urinary tract infections could benefit from narrow-spectrum probabilistic treatment targeting only enterobacteria or streptococci. 

At the hospital level, the development of rapid molecular techniques now allows for rapid de-escalation strategies or even immediate documented treatment, making it possible to spare broad-spectrum molecules [50,51]. Abbara and colleagues showed that an antibiotic stewardship program promoting a strategy of systematic de-escalation associated with restricted use of molecules with a high ecological impact made it possible to achieve a lasting drop in the resistance rate of Pseudomonas aeruginosa in an intensive care unit. In this study, the resistance rate of Pseudomonas aeruginosa to all major classes of antibiotics decreased to less than 20% [26]. The possible generalization of this strategy, facilitated by the availability of narrow-spectrum molecules such as temocillin, should make it possible to recover an epidemiology of resistance similar to that observed in the Netherlands. In fact, in the Netherlands, MRSA and ESBL rates are lower than 5% and almost never warrant the use of carbapenems or glycopeptides.

The indications for broad-spectrum molecules would be reduced to the rare severe septic states with no identified source of infection. 

### 3.3. Supporting Companies to Develop and Produce Targeted Microbiome-Sparing Antibiotics 

By promoting the systematic prescription of narrow-spectrum antibiotics as part of an antimicrobial stewardship program, the indications for narrow-spectrum antibiotics would be very broad, leading to a “profitable” market for the pharmaceutical industry. 

To support the transition from broad-spectrum to narrow-spectrum antibiotics, health authorities could require that any new application for marketing authorization be subject to an assessment of the impact of antimicrobials on the human flora [52]. This approach will help promote narrow-spectrum compounds in the market. 

## 4. Conclusions

The pharmaceutical industry’s main response to the issue of antimicrobial resistance has been to produce new broad-spectrum antibiotics that are still ineffective against multidrug resistant bacteria, thus amplifying antibiotic resistance. In order to combat this phenomenon, antibiotic stewardship programmes aim to restrict the use of these new antibiotics. The development of broad-spectrum antibiotics to control multidrug resistant bacteria is thus an obsolete business model. 

In our opinion, it is now essential to redirect research and development towards new, narrow-spectrum antibiotics with a low ecological impact. These antibiotics could be prescribed for common conditions such as sore throats and cystitis, which account for the bulk of antibiotic consumption in humans. Narrow-spectrum, targeted, microbiome-sparing antibiotics could help control antibiotic resistance while being economically sustainable. Their development and production should be supported by governments, which would ultimately benefit from reduced health costs. 

The pharmaceutical industry must reorient research and development in antibiotic therapy towards the production of narrow-spectrum antibiotics with a low ecological impact.

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
