# Peer review of "The Production of Antibiotics Must Be Reoriented: Repositioning Old Narrow-Spectrum Antibiotics, Developing New Microbiome-Sparing Antibiotics"

_antibiotics, 2022, doi:10.3390/antibiotics11070924_

Round 1

Reviewer 1 Report

The author clearly claims the importance for the industry to switch from developing broad-spectrum antibiotics to narrow-spectrum antibiotics. I think this paper can be helpful to the antibiotics development field.  The only concern is if the author can provide more detailed data or charts to show the benefits that will be stronger.

Author Response

Thank you for your feedback, we have added several paragraphs with examples to illustrate our points. Unfortunately, the overall impact is not measurable and we can only make assumptions.

Reviewer 2 Report

This paper studies “The Pharmaceutical Industry Must Reorient Research and Development in Antibiotic Therapy towards the Production of 3 Narrow-Spectrum Antibiotics with Low Ecological Impact.”- Findings highlight Narrow-spectrum, targeted, microbiome-sparing antibiotics could help control antibiotic resistance while being economically sustainable. Their development and production should be supported by governments which would eventually benefit from lower health-related costs.

This version is not acceptable at all, and I strongly recommend the rejection of this paper. I have not seen the minimum standard of the MDPI journal in this paper. However, I suggest the authors the following comments:

  1. The structure of your paper is so weird. You have not provided even one format of the MDPI journal. You should divide the structure of your paper to:

Introduction, methodology, discussion, and conclusion.

2. You should explain the structure of the paper at the end of the introduction.

3. You just provided 12 references which are so weird as well. You should add at least 20 recent references

4. What is your paper's novelty, research gap, and research questions?

5. Your topic is too long. Remove unnecessary parts.

6. Explain your findings in the abstract.

7. The content is too short, and this paper does not have a coherent structure.

Author Response

  1. The structure of your paper is so weird. You have not provided even one format of the MDPI journal. You should divide the structure of your paper to:

Introduction, methodology, discussion, and conclusion.

Thank you for your comments. We have not respected the guidelines of the authors of the antibiotics journal and we would like to apologize for this. We have modified the structure of the article. We have added the introductory and concluding paragraphs according to similar "perspectives" articles published in the journal

  1. You should explain the structure of the paper at the end of the introduction.

We have added this paragraph at the end of the introduction.

  1. You just provided 12 references which are so weird as well. You should add at least 20 recent references

We have added 39 new references

  1. What is your paper's novelty, research gap, and research questions?

We bring a new vision by proposing to redirect the strategy of antibiotic development. In our opinion, it is now essential to redirect research and development towards new narrow-spectrum antibiotics with low ecological impact. These antibiotics could be prescribed for common pathologies such as sore throats and cystitis, which account for the bulk of antibiotic consumption in humans. Narrow-spectrum, targeted, microbiome-sparing antibiotics could help control antibiotic resistance while being economically sustainable. We do not provide proof but only a new point of view.

  1. Your topic is too long. Remove unnecessary parts.

We have taken your comment into account and the word count is now 2385. The minimum number of words requested by the editor is 2000.

  1. Explain your findings in the abstract.

We have taken your remarks into consideration and we have modified the abstract.

  1. The content is too short, and this paper does not have a coherent structure

We have taken your remarks into consideration and we have modified the structure by adding an introduction and a conclusion with 39 new references.

Reviewer 3 Report

Thank you for the opportunity to review this manuscript. The concept described is a hot topic, since it's about novel antibiotics development considering antibiotic stewardship model and ecological impact.

However, I have several concern before publication of this article:

1) The article needs to undergo marked professional English editing, otherwise it is frequently unclear and misleading.

2) Concepts should be widely explained, and better characterized. Each section should be expanded.

3) There are no references. Authors should check scientific literature to search other similar works or paper about antibiotics and MDR bacteria. I suggest to add these three works:

Erdem H, Hargreaves S, Ankarali H, et al. Managing adult patients with infectious diseases in emergency departments: international ID-IRI study. J Chemother. 2021;33(5):302-318. doi:10.1080/1120009X.2020.1863696;

El-Sokkary R, Uysal S, Erdem H, et al. Profiles of multidrug-resistant organisms among patients with bacteremia in intensive care units: an international ID-IRI survey. Eur J Clin Microbiol Infect Dis. 2021;40(11):2323-2334. doi:10.1007/s10096-021-04288-1

Marino A, Munafò A, Zagami A, et al. Ampicillin Plus Ceftriaxone Regimen against Enterococcus faecalis Endocarditis: A Literature Review. J Clin Med. 2021;10(19):4594. Published 2021 Oct 6. doi:10.3390/jcm10194594

Author Response

1) The article needs to undergo marked professional English editing, otherwise it is frequently unclear and misleading.

Thank you for your comments, we had the article proofread by a native English speaker

2) Concepts should be widely explained, and better characterized. Each section should be expanded.

We have added concrete examples to illustrate our remarks as well as 39 references.

3) There are no references. Authors should check scientific literature to search other similar works or paper about antibiotics and MDR bacteria. I suggest to add these three works:

Thank you for your suggestions, we have added some among the 39 references.

Erdem H, Hargreaves S, Ankarali H, et al. Managing adult patients with infectious diseases in emergency departments: international ID-IRI study. J Chemother. 2021;33(5):302-318. doi:10.1080/1120009X.2020.1863696;

El-Sokkary R, Uysal S, Erdem H, et al. Profiles of multidrug-resistant organisms among patients with bacteremia in intensive care units: an international ID-IRI survey. Eur J Clin Microbiol Infect Dis. 2021;40(11):2323-2334. doi:10.1007/s10096-021-04288-1

Marino A, Munafò A, Zagami A, et al. Ampicillin Plus Ceftriaxone Regimen against Enterococcus faecalis Endocarditis: A Literature Review. J Clin Med. 2021;10(19):4594. Published 2021 Oct 6. doi:10.3390/jcm10194594

Reviewer 4 Report

I have read with interest the manuscript by Diamantis et al., since AMR represents a significant concern, hence new methods to combat this problem are required.

I have some comments to be addressed to improve the manuscript:

1. There is numerous information provided but without references. For example, in the section "A Brief History of Antibiotics and Antibiotic Therapy" there is no reference, even though a lot of data is presented. Please consider adding some supplementary references.

Here are just some examples of information that should be supported with a reference:

  • Research in antibiotic therapy began in the 1940s with a major expansion in the 1980s.
  • For physicians, the failure of historically effective probabilistic treatments such as the use of fluoroquinolones and then cephalosporins in the treatment of urinary tract infections required regular adaptation of recommendations to guarantee patients a failure rate of less than 10%. This was followed by an escalation from fluoroquinolones to 3GCs and, in some countries, to carbapenems, which accelerated the emergence of resistance. 
  • Today, over two thirds of antibiotics prescribed in outpatient care are wrongly given to patients with respiratory viruses.
  • Since the 2000s, bacterial resistance has increased rapidly with the emergence of new resistance enzymes spreading in hospitals but also in the general population with the emergence of CTXM ESBLs and then carabapenemases with NDMs in India and Ox48s in Africa. 
  • In 2020, 46 new antibiotics were in development, including 26 antibiotics active against WHO priority pathogens, 10 of which were β-lactam and β-lactamase inhibitor combinations.
  • Only 20% of antimicrobials entering phase 1 will ever be marketed for medical use.
  • For example, murepavadine (POL7080) is an inhibitor of LptD, a membrane protein involved in surface lipoprotein (SLP) transport, with a very narrow spectrum restricted to Pseudomonas aeruginosa. Afabicin (DEBIO-1450) is an inhibitor of Fab1 involved in the synthesis of membrane fatty acids with a very narrow activity directed exclusively against Staphylococcus aureus. 

2. There is a mix of British and American English. 

3. Row 64 - please correct the word "carabapenemases".

4. Row 52 - please replace "have" (plural) with "has" (singular).

Author Response

Thank you for your constructive comments which we have taken into account in this revised version of the manuscript. We have added the requested references

  1. There is a mix of British and American English. 

Thank you for your comments, we had the article proofread by a native English speaker

  1. Row 64 - please correct the word "carabapenemases".

Thank you for spotting this, it has been corrected.

  1. Row 52 - please replace "have" (plural) with "has" (singular)

This has been corrected.

Round 2

Reviewer 2 Report

This paper studies the pharmaceutical industry must reorient research and development in antibiotic therapy towards the production of narrow-spectrum antibiotics with low ecological impact. This version of your paper is much better. However,  However, there are a few weaknesses that should be addressed in this paper. Therefore, I suggest the authors resubmit it after a minor revision. My suggestions are as follows:

1. Your topic is too long. Please summarize the topic of your paper.

2. Please divide your paper as follows:

- Introduction and literature review 

- Methodology

- Discussion 

-Results

-Conclusion

3. The structure of your paper is weird. Please mention the structure of your paper at the end of the introduction. For example:

The rest of this paper is organized as follows:

Section 1 describes.......Section 2 provides........section 3 proposes 

4. Please include all sections in numbers based on MDPI structure.

5. The main motivation behind this study, the originality of the work (i.e., what are the differences between this work and other published works, what did you emphasize with this work, etc.) should be revealed at the very end paragraph of the manuscript. This section must be realistic, objective, and clearly-written, so that the concerned reader will get the maximum efficiency. 

Author Response

  1. Your topic is too long. Please summarize the topic of your paper.

Thank you for your constructive comments. We have shortened the title

  1. Please divide your paper as follows:

We have divided the article as requested

- Introduction and literature review

- Methodology

- Discussion

-Results

-Conclusion

  1. The structure of your paper is weird. Please mention the structure of your paper at the end of the introduction. For example:

The rest of this paper is organized as follows:

We have removed sections as requested

Section 1 describes.......Section 2 provides........section 3 proposes

  1. Please include all sections in numbers based on MDPI structure.

  1. The main motivation behind this study, the originality of the work (i.e., what are the differences between this work and other published works, what did you emphasize with this work, etc.) should be revealed at the very end paragraph of the manuscript. This section must be realistic, objective, and clearly-written, so that the concerned reader will get the maximum efficiency.

We have added a concluding sentence at the very end of the article as requested.

Reviewer 3 Report

Authors have improved the manuscript which, in my opinion, deserves publication after minor english checks and format check.

References list has been improved too, however I think it would be useful to add the reference I have previously suggested.

Author Response

Authors have improved the manuscript which, in my opinion, deserves publication after minor english checks and format check.

References list has been improved too, however I think it would be useful to add the reference I have previously suggested.

Thank you for your comments we have added the requested reference El sokkari et al EJCMID 2021

Reviewer 4 Report

I appreciate the authors' effort in addressing the comments and suggestions, and I accept the modifications and responses from the authors. 

I find this second version correct. 

Author Response

Thank you for your appreciation